# Telomere shortening and the transition to family caregiving in the Reasons for Geographic and Racial Differences in Stroke (REGARDS) study

Nicole D. Armstrong[1], Marguerite R. Irvin[1], William E. Haley[2], Marcela D. Blinka[3], Debora Kamin Mukaz[4], Amit Patki[5], Sue Rutherford Siegel[6], Idan Shalev[6], Peter Durda[7], Rasika A. Mathias[8], Jeremy D. Walston[3], David L. Roth[3]*

1 Department of Epidemiology, University of Alabama at Birmingham, Birmingham, AL, United States of America, 2 School of Aging Studies, University of South Florida, Tampa, FL, United States of America, 3 Center on Aging and Health, Division of Geriatric Medicine and Gerontology, Johns Hopkins University, Baltimore, MD, United States of America, 4 Department of Medicine, Larner College of Medicine, University of Vermont, Burlington, VT, United States of America, 5 Department of Biostatistics, University of Alabama at Birmingham, Birmingham, AL, United States of America, 6 Department of Biobehavioral Health, Pennsylvania State University, University Park, PA, United States of America, 7 Department of Pathology and Laboratory Medicine, Larner College of Medicine, University of Vermont, Burlington, VT, United States of America, 8 Division of Allergy and Clinical Immunology, Department of Medicine, Johns Hopkins School of Medicine, Baltimore, MD, United States of America

* droth@jhu.edu

**Data Availability Statement:** In cooperation with the Institutional Review Board (IRB) of the University of Alabama at Birmingham, the

## Abstract

Telomere length (TL) is widely studied as a possible biomarker for stress-related cellular aging and decreased longevity. There have been conflicting findings about the relationship between family caregiving stress and TL. Several initial cross-sectional studies have found associations between longer duration of caregiving or perceived stressfulness of caregiving and shortened TL, suggesting that caregiving poses grave risks to health. Previous reviews have suggested the need for longitudinal methods to investigate this topic. This study examined the association between the transition to family caregiving and change in TL across ~9 years. Data was utilized from the Caregiving Transitions Study, an ancillary study to the Reasons for Geographic and Racial Differences in Stroke (REGARDS) study. TL was assayed using qPCR and analyzed as the telomere-to-single copy gene ratio for each participant at baseline and follow-up. General linear models examined the association between caregiving status and the change in TL for 208 incident caregivers and 205 controls, as well as associations between perceived stress and TL among caregivers. No association was found between TL change and caregiving ($p = 0.494$), and fully adjusted models controlling for health and socioeconomic factors did not change the null relationship ($p = 0.305$). Among caregivers, no association was found between perceived caregiving stress and change in TL ($p = 0.336$). In contrast to earlier cross-sectional studies, this longitudinal, population-based study did not detect a significant relationship between the transition into a family caregiving role and changes in TL over time. Given the widespread citation of previous findings suggesting that caregiving shortens telomeres and places caregivers at risk of

REGARDS project facilitates data sharing through formal data use agreements. Investigators who wish to access the data and code for these analyses should send their requests to regardsadmin@uab.edu.

**Funding:** This work was supported by the cooperative agreement U01 NS041588 co-funded by the National Institute of Neurological Disorders and Stroke (NINDS) and the National Institute on Aging (NIA), National Institutes of Health, and Department of Health and Human Service. The Caregiving Transitions Study was further supported by an investigator-initiated grant (RF1 AG050609) from the NIA. Additional support was provided by the Johns Hopkins University Claude D. Pepper Older Americans Independence Center funded by the NIA (P30 AG021334). The content is solely the responsibility of the authors and does not necessarily represent the official views of the NINDS or the NIA. Representatives of the NINDS were involved in the review of the manuscript but were not directly involved in the collection, management, analysis, or interpretation of the data.

**Competing interests:** The authors have declared that no competing interests exist.

early mortality, these results demonstrate the potential need of a more balanced narrative about caregiving.

## Introduction

An increase in life expectancy and an aging population has resulted in increased risk and prevalence of age-related diseases including autoimmune disorders, cardiovascular diseases, and dementia [1]. Chronic stress has been widely studied as a potential risk factor for a variety of illnesses [2, 3] and for reduced longevity [4, 5]. Because caring for a family member with a chronic illness or disability can be a lengthy and highly stressful experience, family caregiving has been viewed as an important paradigm to study the associations among stress, illness, biomarkers, and longevity. A widely-cited study reported a 63% increase in mortality among spousal caregivers who reported strain compared to spouses who were not caregivers [6]. This early mortality finding served as a major scientific foundation for several subsequent studies of possible biological mechanisms for the negative effects of caregiving on health and mortality. Studies of immunity and inflammatory biomarkers, as well as telomere length (TL), appeared promising due to their association with the rates of chronic disease and premature mortality [7, 8]. Research on caregiving stress is widely viewed as one important focus in examining chronic psychosocial stress and health outcomes, along with studies of early life adversity, childhood abuse, stressful life events, work-related stressors, and financial strain [9].

Telomeres, the protective caps on the ends of chromosomes, play a key role in chromosomal stability. When telomeres become critically short, cells enter a state of replicative arrest, or senescence, which is postulated to be a core driver of age-related decline and one of the hallmarks of aging [10–12]. Previous studies have suggested an average TL attenuation of ~25 bp/year in adults [13], longer TL in women compared to men [14, 15], and longer TL in African ancestry populations [15, 16]. Furthermore, interventions such as physical activity and diet, particularly in older adults, can help slow age-related reductions in TL [17–20]. A number of cross-sectional studies have also shown that factors, such as chronic stress, are associated with shorter TL [21–24], although a recent systematic review and meta-analysis concluded that measures of perceived stress are only minimally associated with small decreases in TL (r = -0.06) [21]. While there are few longitudinal studies examining changes in stressful life events and TL, one recent study found that increases in self-reported stressful life events over time were associated with subsequent declines in TL among monozygotic, but not dizygotic, twins [25].

Family caregivers of older adults with dementia and other conditions often report high levels of stress and have been studied as a group that might experience telomere shortening; however, previous findings on caregiving and TL are mixed. Three earlier cross-sectional studies have reported a relationship between shorter TL and being a caregiver [12, 26], or between shorter TL and longer duration of caregiving and higher perceived stress among caregivers [27]. However, potential weaknesses of these initial studies included the use of cross-sectional designs, caregiving sample sizes of 58 or fewer, and that both caregivers and non-caregiving controls were recruited from convenience samples (e.g., clinics and support groups for caregivers, volunteers for non-caregivers). Notably, convenience samples could lead to selection biases since caregivers recruited from convenience samples typically report more distress than those recruited via population-based approaches [28, 29].

More recent studies of caregiving and TL based in more general populations with larger sample sizes have reported inconsistent results. One such study with 240 caregivers and 98

non-caregiving controls reported no significant association between caregiving and TL but did find subgroups (caregivers providing more hours of care, caring for younger people, or greater strain) with shorter TL [30]. A population-based study of 563 caregivers and 627 non-caregiving controls found no significant associations between long-term patterns of caregiving intensity and change in TL after an eight-year follow-up [31]. The most recently published study examining the caregiving-TL relationship from the Cebu Longitudinal Health and Nutrition Survey also observed no evidence of an association between chronicity of care and TL and suggested that previous significant results linking caregiving and TL could be limited to certain types of caregiving or be statistical artifacts due to small sample sizes [32].

Given these mixed findings, a recent review suggested that future research on psychosocial stress and TL focus on longitudinal studies in population-based samples [9]. While the Nurses' Health Study [31] examined changes in TL over an 11-year period, the first assessment of TL in that study occurred after many of the participants were already caregivers. To the best of our knowledge, there are no previous studies that have examined how the caregiving transition (compared to non-caregiving) influences prospective changes in TL within the same individuals.

The Caregiving Transitions Study (CTS) is an ancillary study to Reasons for Geographic and Racial Differences in Stroke (REGARDS) cohort study. The CTS selected cases who became family caregivers during follow-up and non-incident caregiving controls leveraging longitudinal samples and measures of psychological distress. This design enabled examination of whether the transition to incident caregiving was associated with greater changes in TL. We further examined whether perceived caregiving stress was associated with TL within the caregiving case group, and finally, whether TL was associated with a more general measure of perceived stress in the full CTS. We hypothesized that the transition to caregiving would constitute a form of chronic stress and that the transition to caregiving would be accompanied by a reduction in TL. The rich data resource paired with banked samples in REGARDS allowed the current study to address limitations of previous research yielding conflicting findings on the relationship of caregiving stress and TL changes.

## Methods

### The REGARDS study

The REGARDS study is a national longitudinal study that enrolled 30,239 adults aged 45 or older in 2003–2007. Participants were oversampled by design if they were residents of the "stroke belt" region of the Southeastern United States or Black. Participants completed a computer-assisted telephone interview (CATI) and an in-home visit where blood and urine samples were collected at baseline (visit 1), as well as a follow-up CATI and in-home visit ~9 years later (visit 2). Additional detail on the REGARDS study have been described elsewhere [33].

### Caregiver Transitions Study (CTS) participants

The CTS is a nested case-control study within REGARDS that enrolled REGARDS participants who assumed a role as a family caregiver between the first and second REGARDS in-home assessment. Additional details used to screen and enroll participants in the CTS have been described previously [34]. Briefly, during the REGARDS baseline CATI, participants were asked: "Are you currently providing care on an on-going basis to a family member with a chronic illness or disability? This would include any kind of help such as watching your family member, dressing or bathing this person, arranging care, or providing transportation?" Individuals who answered "no" were categorized as baseline non-caregivers. Approximately 12 years after the first in-home visit, a similar question was asked during a follow-up CATI to

obtain updated caregiving status. Participants who answered "yes" to this question and "no" to the baseline CATI question were screened for incident caregiver enrollment eligibility. Those who answered "no" to both questions during the separate CATIs were eligible to serve as non-caregiving controls. During the caregiving screening interview, each potential incident caregiver reported the year and month that they started aiding the care recipient because of his or her health problem. To be eligible, the onset of caregiving had to be at least 6 months after the first REGARDS in-home assessment and blood draw and at least 3 months before the second REGARDS in-home assessment and blood draw [34, 35]. For the incident caregivers who were enrolled in the CTS, the reported date when caregiving began due to the care recipient's need for assistance with activities of daily living was 3.4 years (SD = 2.4), on average, before the second REGARDS in-home assessment. Once an eligible incident caregiver was enrolled, a list of individuals who matched that caregiver on seven factors from the REGARDS baseline CATI including age within 5 years, sex, race, education level, marital status, self-rated health, and self-reported history of cardiovascular disease was assembled. These persons were contacted by telephone and screened for possible enrollment as non-caregiving controls in the CTS [34].

In the present telomere study, participants were included in analyses if they were enrolled in the CTS and had both baseline (visit 1) and follow-up (visit 2) telomere measures, resulting in a total of 413 participants (208 incident caregivers and 205 non-caregiving controls). The REGARDS study and the CTS were both reviewed and approved by institutional review boards of the University of Alabama at Birmingham and Johns Hopkins University. Written informed consent was obtained from all participants prior to the first blood sample.

## Covariates

The baseline (visit 1) characteristics included as covariates in general linear models were reported age, race/ethnicity (non-Hispanic Black or non-Hispanic White), annual income (less than $20,000/refused, $20,000-$34,000, $35,000-$74,000, or $75,000 and above), educational attainment (≤high school graduate, some college, or college graduate and above), and relationship status (married, single/other, divorced, or widowed). Baseline lifestyle and health factors included information on cigarette smoking (current, past, or never), alcohol consumption (current, past, or never), and baseline body mass index (BMI, kg/m$^2$). Perceived stress was measured during the follow-up CATI with the 4-item short form of the Cohen Perceived Stress Scale (PSS) [36]. The PSS is a widely-used and previously validated [37] measure of the degree to which participants evaluate the events and circumstance of their lives as being unpredictable, uncontrollable, overloaded, or beyond their ability to manage [38]. Reponses to these items were summed to create a total score, ranging from 0–16 (the current study range: 0–13) and were analyzed as both a continuous variable and as tertiles (low, moderate, or high perceived stress), as previously described in the CTS [38] and REGARDS [39].

## Telomere length assay

The detailed descriptions of sample handling and processing, as well as details regarding the qPCR assay and quality control, are summarized in S1 Table in accordance with guidelines recommended by the Telomere Research Network (https://trn.tulane.edu). Briefly, TL assays were conducted using stored DNA extracted from blood at the Pennsylvania State University College of Health and Human Development Biomarker Core Lab (University Park, PA). DNA concentrations were determined by the ThermoFisher Scientific Quant-iT dsDNA Assay kit (ThermoFisher Scientific, Waltham, MA) using the manufacturers protocol. TL was determined using a modified qPCR assay developed by Cawthon [40]. A separate standard curve for a single-copy gene was also performed that incorporates an 82bp duplex oligomer for the

Interferon beta 1 (*IFNB1*) gene. Calculation of total TL per diploid genome for each sample was performed by dividing the number of diploid genomes in the sample into the average length determined from the telomere standard curve for each sample.

PCR amplifications were established using the QIAgility robotic workstation (Qiagen, Germany). Real-time qPCR was performed using a Rotor-Gene Q 5plex HRM (Qiagen). The Rotor-Gene Q also determined the threshold Ct values for both the telomere and *IFNB1* oligomer standard curves for each plate. Each sample was run in triplicate and distribution of samples in the 100-well plate were blinded and randomized. DNA from a long telomere positive control (cell line 1301; accession number 01051619, European Collection of Cell Cultures, UK Equipment) was also included on each plate. To mitigate batch effects, at least 10 repeats from reference controls were included with each run. To assure plate-to-plate consistency, the median Ct threshold was determined across all plates for this cell line and used for each plate. The interplate and intraplate coefficient of variation (± standard deviation, SD) of the reference control was 13.8% (±0.9) and the 4.4% (± 0.3), respectively. Importantly, both samples (visit 1 and visit 2) from each individual were run on the same plate.

The median telomere value and the median *IFNB1* value were used to determine the telomere-to-single copy gene ratio (T/S) for each sample. The T/S ratios (referred to as TL herein) were calculated using the following equation:

$$2^{-\Delta Ct} = \left[ \frac{2^{Ct(telomere)}}{2^{Ct(IFNB1)}} \right]^{-1}$$

where $C_t^{(telomere)}$ is the cycle number to reach the PCR threshold for the telomere sample and $C_t^{(IFNB1)}$ is the cycle number to reach the PCR threshold for the single copy gene, *IFNB1*.

## Statistical analysis

Frequency distributions of TL (assessed as the T/S ratio) were examined at both the first and second visits. These distributions were observed to be positively skewed and the natural logarithm of the TL, ln(T/S), was used in subsequent analysis to account for skewness. The dependent variable in regression models was the change of TL over time and was defined as $\Delta$(T/S) = ln(V$_2$)–ln(V$_1$), where V$_2$ and V$_1$ represent the T/S values at visit 2 and visit 1, respectively. This is equivalent to $\Delta$(T/S) = ln(V$_2$/V$_1$) and was interpreted as a fold-change of TL ratios and not the absolute difference between the two time points.

We used Tukey's interquartile range (IQR) method to detect TL outliers at the first and second visits separately using the natural logarithm transformed data. Briefly, any values that were less than Q1–3$^*$ (Q3-Q1) or more than Q3 + 3$^*$(Q3-Q1) were designated extreme outliers and were set as missing. Using this method, we recorded one low value and two high values as outliers for the first visit.

Descriptive analyses comparing caregiver and non-caregiver characteristics were performed using $\chi^2$ tests or *t* tests and all tests of significance were two-sided. We utilized the method of least squares to fit general linear models controlling for sex, race, visit 1 age, and visit 1 TL and examined the association between caregiving status (exposure) and the $\Delta$(T/S) (outcome) in a base model (Model 1). Additional models adjusted for cigarette smoking, alcohol use, and BMI (model 2), as well as income, educational attainment, and relationship status (model 3). Secondary models performed within the caregiving sample adjusted for sex, race, visit 1 age, and visit 1 TL length, and examined the associations between caregiving duration and perceived stress at visit 2 and $\Delta$(T/S). Bivariate correlations were performed between measures of TL, age, and perceived stress. Statistical significance was set at $p < 0.05$ for all analyses. All analyses were conducted in SAS version 9.4 (SAS Institute Inc., Cary, NC).

## Results

Descriptive information for the incident caregivers and non-caregiving controls are presented in Table 1. The demographic and lifestyle baseline factors did not differ between caregivers and controls except for BMI, where the mean (±SD) caregiver BMI was 30.66 (±6.43) versus a mean control BMI of 28.74 (±6.23) ($p$ = 0.002). The mean visit 1 TL (on the natural log scale) for caregivers was 5.95 (±0.99) and 6.15 (±1.05) for non-caregiving controls ($p$ = 0.049), while the mean visit 2 TL (on the natural log scale) was 5.95 (±0.99) and 6.00 (±1.03) for caregivers and controls, respectively ($p$ = 0.588). TL at visit 1 and visit 2 were significantly correlated (r(411) = 0.56, $p<0.001$), which indicated telomere stability over time. We also observed that change in TL was significantly correlated (r(411) = -0.48, $p<0.001$) with visit 1 TL (S2 Table, S1 Fig). The average percent $\Delta$(T/S) per year follow-up for caregivers was 0.03%, while the average percent $\Delta$(T/S) per year follow-up for controls was -1.55% on the natural log scale. When examining the cross-sectional relationships between TL, age at visits 1 and 2, or perceived stress at visit 2, we did not observe any correlation between these variables in our data (S2 Table).

Table 2 presents results for the three models examining the relationship between incident caregiving status and change in TL. When adjusting for age at visit 1, sex, race, and TL at visit 1, we did not observe a significant association between the change in TL and caregiving status, where caregivers had a nominal 1.06-fold increase in TL compared to controls ($\beta$ = 0.057, SE = 0.08, $p$ = 0.494). Upon further adjustment of BMI, cigarette smoking, and alcohol use (model 2; $\beta$ = 0.079, SE = 0.09, $p$ = 0.352), as well as income, educational attainment, and relationship status (model 3; $\beta$ = 0.089, SE = 0.09 $p$ = 0.305), the null relationship between TL change and caregiving status was unchanged.

Secondary models on the caregiving sample, adjusting for age, sex, race, and visit 1 TL, showed change of TL was not significantly associated with perceived stress at time 2 or with caregiving duration. Results were similar in a more fully adjusted model (Table 3). Evaluating the change in telomere length across low, moderate, and high perceived stress categories did not yield statistically significant differences across the entire CTS telomere subset, nor in the caregivers (S3 Table).

## Discussion

In the current study, we examined the relationship between the transition to family caregiving and the change in TL over a period of ~9 years, including on average, more than 3 years after the caregivers transitioned into sustained and extensive caregiving roles. Previous studies on TL and caregiving have yielded mixed and inconsistent results. Our longitudinal data from the REGARDS population-based study suggest that caregivers, as a general group, do not show significant change in TL or telomeric shortening over the first few years of caregiving compared to similar participants who are not family caregivers. Our lack of TL differences between caregivers and controls is consistent with other recent population-based studies [30–32] and does not support earlier cross-sectional studies that found shorter telomeres among caregivers compared to non-caregiving comparison samples [12, 26]. Our study is unique in examining changes in TL from before to after the onset of caregiving among the sample of incident caregivers versus controls in our analysis. Since one previous study [27] reported that longer duration of caregiving and higher perceived stress was associated with shorter telomere length among caregivers, we also conducted analyses within the caregiving sample to assess the relationship of these factors to TL. We found no significant associations.

The heightened mortality of caregivers compared to non-caregivers in one initial study [6] has not been replicated in several subsequent studies that have found the opposite pattern;

**Table 1. Descriptive characteristics for incident caregivers and non-caregiving controls.**

| | Caregivers ($n$ = 208) | Controls ($n$ = 205) | P[a] |
|---|---|---|---|
| **Age at visit 1 (years)[b]** | 59.84 ± 7.64 | 59.42 ± 7.15 | 0.558 |
| **Sex** | | | |
| Female | 136 (65.38%) | 138 (67.32%) | 0.678 |
| Male | 72 (34.62%) | 67 (32.68%) | |
| **Race** | | | |
| Black | 72 (34.62%) | 71(34.63%) | 0.997 |
| White | 136 (65.38%) | 134 (65.37%) | |
| **BMI at visit 1 (kg/m²)[c]** | 30.66 ± 6.43 | 28.74 ± 6.23 | 0.002 |
| **Income** | | | |
| < $20k/Refused | 36 (17.31%) | 33 (16.10%) | 0.348 |
| $20-$34k | 43 (20.67%) | 38 (18.54%) | |
| $35-$74k | 87 (41.83%) | 77 (37.56%) | |
| ≥ $75k | 42 (20.19%) | 57 (27.80%) | |
| **Educational Attainment** | | | |
| ≤HS graduate | 60 (28.85%) | 55 (26.83%) | 0.286 |
| Some college | 65 (31.25%) | 53 (25.85%) | |
| ≥College graduate | 83 (39.90%) | 97 (47.32%) | |
| **Alcohol Use** | | | |
| Current | 121 (58.17%) | 120 (58.54) | 0.754 |
| Never | 60 (28.85%) | 63 (30.73%) | |
| Past | 27 (12.98%) | 22 (10.73%) | |
| **Cigarette Use** | | | |
| Current | 21 (10.10%) | 14 (6.83%) | 0.358 |
| Never | 111 (53.37%) | 121 (59.02%) | |
| Past | 76 (36.54%) | 70 (34.15%) | |
| **Relationship Status** | | | |
| Single/Other | 19 (9.13%) | 14 (6.83%) | 0.650 |
| Divorced | 26 (12.50%) | 25 (12.20%) | |
| Widowed | 13 (6.25%) | 9 (4.39%) | |
| Married | 150 (72.12%) | 157 (76.59%) | |
| **PSS, Visit 2[c]** | 3.06 ± 2.64 | 2.47 ± 2.64 | 0.026 |
| **PSS, Visit 2 Tertile[c]** | | | |
| Low (0–1) | 71 (35.15%) | 94 (45.85%) | 0.008 |
| Moderate (2–4) | 69 (34.16%) | 74 (36.10%) | |
| High (5+) | 62 (30.69%) | 37 (18.05%) | |
| **Time between visits, years** | 9.30 ± 0.96 | 9.28 ± 0.81 | 0.863 |
| **LN T/S, Visit 1** | 5.95 ± 0.99 | 6.15 ± 1.05 | 0.049 |
| **LN T/S, Visit 2** | 5.95 ± 0.99 | 6.00 ± 1.03 | 0.588 |
| **ΔT/S** | 0.00 ± 0.91 | -0.14 ± 1.00 | 0.127 |

[a] Statistical significance set at p<0.05.

[b] Continuous traits are described as mean ± SD; categorical traits are described as N (%).

[c] Participants missing data for BMI: n = 1 caregiver, 1 control; PSS, visit 2: n = 6 caregivers

**Abbreviations:** SD- standard error; BMI- body mass index; PSS- Perceived Stress Scale; LN-natural log; T/S- telomere to single copy gene ratio; ΔT/S- change in telomere ratio (visit 2- visit 1).

**Table 2. Effects of caregiving status on the change in TL (ΔT/S).**

| | Model 1 | | Model 2 | | Model 3 | |
|---|---|---|---|---|---|---|
| | β (SE)[a] | P[b] | β (SE) | P | β (SE) | P |
| Intercept | 2.600 (0.44) | <0.001 | 2.902 (0.53) | <0.001 | 2.766 (0.58) | <0.001 |
| **Caregiving Status** | | | | | | |
| Control | Ref. | | Ref. | | Ref. | |
| Caregiver | 0.057 (0.08) | 0.494 | 0.079 (0.09) | 0.352 | 0.089 (0.09) | 0.305 |
| **Visit 1 TL** | -0.448 (0.04) | <0.001 | -0.452 (0.04) | <0.001 | -0.459 (0.04) | <0.001 |
| **Visit 1 Age** | -0.001 (0.01) | 0.906 | -0.002 (0.01) | 0.761 | 0.001 (0.01) | 0.862 |
| **Sex** | | | | | | |
| Male | Ref. | | Ref. | | Ref. | |
| Female | 0.086 (0.09) | 0.348 | 0.065 (0.10) | 0.495 | 0.084 (0.10) | 0.390 |
| **Race** | | | | | | |
| White | Ref. | | Ref. | | Ref. | |
| Black | -0.012 (0.09) | 0.892 | 0.012 (0.09) | 0.898 | 0.041 (0.10) | 0.671 |
| **Visit 1 BMI** | -- | -- | -0.006 (0.01) | 0.425 | -0.004 (0.01) | 0.534 |
| **Cigarette Smoking** | | | | | | |
| Never | -- | -- | Ref. | | Ref. | |
| Current | -- | -- | -0.169 (0.16) | 0.282 | -0.174 (0.16) | 0.278 |
| Past | -- | -- | -0.091 (0.09) | 0.338 | -0.094 (0.10) | 0.325 |
| **Alcohol Use** | | | | | | |
| Never | -- | -- | Ref. | | Ref. | |
| Current | -- | -- | 0.013 (0.10) | 0.899 | 0.017 (0.10) | 0.864 |
| Past | -- | -- | -0.107 (.015) | 0.469 | -0.107 (0.15) | 0.476 |
| **Educational Attainment** | | | | | | |
| ≤High school graduate | -- | -- | -- | -- | Ref. | |
| Some college | -- | -- | -- | -- | -0.135 (0.12) | 0.248 |
| ≥College graduate | -- | -- | -- | -- | -0.107 (0.11) | 0. 331 |
| **Income** | | | | | | |
| <$20k/refused | -- | -- | -- | -- | Ref. | |
| $20-$34k | -- | -- | -- | -- | -0.023 (0.14) | 0.870 |
| $35k-$74k | -- | -- | -- | -- | 0.073 (0.13) | 0. 562 |
| ≥$75k | -- | -- | -- | -- | 0.154 (0.14) | 0. 288 |
| **Relationship Status** | | | | | | |
| Single/Other | -- | -- | -- | -- | Ref. | |
| Divorced | -- | -- | -- | -- | -0.041 (0.20) | 0.835 |
| Widowed | -- | -- | -- | -- | -0.272 (0.25) | 0.268 |
| Married | -- | -- | -- | -- | -0.020 (0.17) | 0.908 |

[a] Beta coefficient and SE are based on natural log ratios

[b] Statistical significance set at p<0.05

**Abbreviations**: SE- standard error; T/S- telomere to single copy gene ratio; Ref- reference group; TL-telomere length; BMI- body mass index (kg/m$^2$)

increased longevity among caregivers [28, 41]. Similarly, initial studies of small convenience samples suggested markedly increased inflammation in caregivers [42], but several more recent studies have detected only minimal elevations in inflammation among caregivers, and no significant effects have been detected in most population-based studies [35, 43]. The findings in the caregiving literature on mortality, inflammation, and TL seem to show a pattern that has been identified more broadly in multiple scientific literatures, where small, early studies with dramatic effects are often heavily cited and highly influential even when subsequent

**Table 3. Effects of PSS and caregiving duration on ΔT/S in caregivers.**

| | Model 1 | | Model 2 | |
|---|---|---|---|---|
| | [a]β (SE) | [b]P | [a]β (SE) | [b]P |
| Intercept | 3.056 (0.61) | <0.001 | 3.419 (0.87) | <0.001 |
| Visit 2 PSS | -0.021 (0.02) | 0.336 | -0.022 (0.02) | 0.354 |
| Caregiving Duration (year) | 0.020 (0.02) | 0.412 | 0.025 (0.03) | 0.330 |
| Visit 1 TL | -0.427 (0.06) | <0.001 | -0.433 (0.06) | <0.001 |
| Visit 1 Age | -0.008 (0.01) | 0.285 | -0.007 (0.01) | 0.432 |
| **Sex** | | | | |
| Male | Ref. | | Ref. | |
| Female | -0.068 (0.13) | 0.601 | -0.057 (0.14) | 0.683 |
| **Race** | | | | |
| White | Ref. | | Ref. | |
| Black | 0.111 (0.13) | 0.377 | 0.204 (0.14) | 0.137 |
| **Visit 1 BMI** | -- | -- | -0.010 (0.01) | 0.372 |
| **Cigarette Smoking** | | | | |
| Never | -- | -- | Ref. | |
| Current | -- | -- | -0.324 (0.21) | 0.119 |
| Past | -- | -- | -0.084 (0.13) | 0.534 |
| **Alcohol Use** | | | | |
| Never | -- | -- | Ref. | |
| Current | -- | -- | -0.027 (0.15) | 0.856 |
| Past | -- | -- | -0.273 (0.20) | 0.181 |
| **Educational Attainment** | | | | |
| ≤High school graduate | -- | -- | Ref. | |
| Some college | -- | -- | 0.011 (0.16) | 0.943 |
| ≥College graduate | -- | -- | -0.079 (0.16) | 0.616 |
| **Income** | | | | |
| <$20k/refused | -- | -- | Ref. | |
| $20-$34k | -- | -- | -0.198 (0.20) | 0.317 |
| $35k-$74k | -- | -- | -0.212 (0.17) | 0.227 |
| ≥$75k | -- | -- | 0.065 (0.21) | 0.759 |
| **Relationship Status** | | | | |
| Single/Other | -- | -- | Ref. | |
| Divorced | -- | -- | 0.049 (0.28) | 0.858 |
| Widowed | -- | -- | -0.271 (0.33) | 0.417 |
| Married | -- | -- | 0.159 (0.23) | 0.495 |

[a] Beta coefficient and SE are based on natural log ratios.

[b] Statistical significance set at p<0.05

**Abbreviations**: PSS- Perceived Stress Scale; SE- standard error; T/S- telomere to single copy gene ratio; Ref- reference group; TL-telomere length; BMI- body mass index (kg/m$^2$)

larger studies often fail to replicate those initial results [44]. Widespread, negative information about caregiving that has not been substantiated has the potential to discourage many family members from taking on caregiving responsibilities, or unnecessarily alarming them over exaggerated risks to their own health.

The recent null results on telomere shortening among caregivers from this study and others [30–32] should be considered in the broader context of other recent challenges to the common narrative that caregivers are at heightened risk for major negative health outcomes. It has been

hypothesized that a major stress exposure, such as caregiving, can lead to chronically high levels of perceived stress and/or psychological stress arousal leading to a reduction in telomere maintenance and thus a reduction in TL [21, 45]. Previous studies have explored the relationship between TL and health, and have found shorter TL to be a risk factor for diseases of aging such as cancer [46] and cardiometabolic dysfunction [21, 47]. In the current study, while we observed differences in the degree of perceived stress between caregivers and controls, we did not find any association with perceived stress or the caregiving transition and changes in TL among all ~400 participants or in the caregiving subpopulation.

Another major challenge in this area of research on TL pertains to issues surrounding the measurement of TL itself in the context of a longitudinal design. As with our work, many of the prior studies rely on qPCR-based measurements in whole blood, with only a few exceptions using the less common, laborious Southern blot (SB) approaches. Nettle et al has recently shown that even where cross-sectional correlations between SB and qPCR at baseline and follow-up visits are high, the correlation between SB and qPCR for the change in leukocyte TL between the two visits can be considerably lower due to errors introduced at each time point in the qPCR assay [48]. In our study, intraplate coefficient of variation (± SD) of the reference control was 4.4% (± 0.3) suggesting high quality of our assay especially as all samples per person were run within a single plate (i.e., visit 1 and visit 2 samples were on the same plate). Nonetheless, despite the high-quality data, we do recognize the limitations implicated with qPCR as a tool for examining longitudinal changes in TL. We observed minimal correlations between age and cross-sectional TL or the change in TL. This may be due to smaller age-related effects compared to the variability in qPCR. Furthermore, REGARDS enrolled participants over the age of 45 in a population-based study, and therefore, more extreme telomere changes due to stress could be observed in younger individuals or individuals within a clinical setting.

The present study has several strengths. We selected participants from a large, population-based, national study. Compared to studies of convenience samples, our study should improve generalizability of findings. Second, REGARDS is well-phenotyped, allowing for inclusion of several potential socioeconomic and/or lifestyle factors previously associated with disease and/or chronic stress [49, 50]. Last, the longitudinal nature of this study with ~9-year follow-up range describes the long-term impact of incident caregiving on cellular aging and TL. Limitations to our study include the self-report nature of the caregiving variables and the other behavior measures used as covariates. Thus, reporting biases could potentially influence our results. In addition, the second blood sample was taken an average of 3.4 years after the caregivers transitioned into the caregiving role, and longer durations of caregiving may show a more dramatic impact on TL and other biological sequalae of chronic caregiving stress.

In conclusion, this study provides important evidence from a longitudinal study that fails to support any meaningful association between a transition to family caregiving and telomere shortening over the first 3+ years of the caregiving experience. Future work on the TL over time could focus on how caregiving strain, chronicity, or the type of caregiving (e.g., to a parent, spouse, or child; dementia/cognitive impairment or physical impairment caregiver) might be associated with TL. Ultimately, more research to address the complex relationship between caregiving and measures of cellular aging is needed to improve the well-being of caregivers and care recipients.

## Supporting information

**S1 Table. Telomere Research Network reporting guidelines.**
(DOCX)

**S2 Table. Bivariate correlations between measures of telomere length, age, and perceived stress.**
(DOCX)

**S3 Table. The change in telomere length (ΔT/S) across perceived stress scale categories.**
(DOCX)

**S1 Fig. The distribution of telomere length.** Left panel: ln(T/S) at baseline (visit 1) and right panel: the change in ln(T/S) over the follow-up period.
(DOCX)

## Acknowledgments

The authors thank the other investigators, the staff, and the participants of the REGARDS study for their valuable contributions. A full list of participating REGARDS investigators and institutions can be found at http://www.regardsstudy.org.

## Author Contributions

**Conceptualization:** William E. Haley, Amit Patki, Jeremy D. Walston, David L. Roth.

**Data curation:** Sue Rutherford Siegel, Idan Shalev, Peter Durda.

**Formal analysis:** Nicole D. Armstrong, Amit Patki, Rasika A. Mathias.

**Funding acquisition:** Jeremy D. Walston, David L. Roth.

**Investigation:** Nicole D. Armstrong, Marguerite R. Irvin, Debora Kamin Mukaz.

**Methodology:** Nicole D. Armstrong, Marguerite R. Irvin, William E. Haley, Rasika A. Mathias, David L. Roth.

**Project administration:** David L. Roth.

**Resources:** Peter Durda.

**Supervision:** David L. Roth.

**Writing – original draft:** Nicole D. Armstrong, William E. Haley.

**Writing – review & editing:** Marguerite R. Irvin, Marcela D. Blinka, Debora Kamin Mukaz, Sue Rutherford Siegel, Idan Shalev, Rasika A. Mathias, Jeremy D. Walston, David L. Roth.

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
