## [Decision Letter · Decision Letter 0]

8 Mar 2022

PONE-D-22-03368Telomere Shortening and the Transition to Family Caregiving in the Reasons for Geographic and Racial Differences in Stroke (REGARDS) StudyPLOS ONE

Dear Dr. Roth,

Thank you for submitting your manuscript to PLOS ONE. After careful consideration, we feel that it has merit but does not fully meet PLOS ONE’s publication criteria as it currently stands. Therefore, we invite you to submit a revised version of the manuscript that addresses the points raised during the review process. 

We look forward to receiving your revised manuscript.

Kind regards,

Jian Gu

Academic Editor

PLOS ONE

Journal Requirements:

[This work was supported by the cooperative agreement U01 NS041588 co-funded by the National Institute of Neurological Disorders and Stroke (NINDS) and the National Institute on Aging (NIA), National Institutes of Health, and Department of Health and Human Service. The Caregiving Transitions Study was further supported by an investigator-initiated grant (RF1 AG050609) from the NIA. Additional support was provided by the Johns Hopkins University Claude D. Pepper Older Americans Independence Center funded by the NIA (P30 AG021334). The content is solely the responsibility of the authors and does not necessarily represent the official views of the NINDS or the NIA.  Representatives of the NINDS were involved in the review of the manuscript but were not directly involved in the collection, management, analysis, or interpretation of the data.The authors thank the other investigators, the staff, and the participants of the REGARDS study for their valuable contributions.  A full list of participating REGARDS investigators and institutions can be found at http://www.regardsstudy.org.]

 [This work was supported by the cooperative agreement U01 NS041588 co-funded by the National Institute of Neurological Disorders and Stroke (NINDS) and the National Institute on Aging (NIA), National Institutes of Health, and Department of Health and Human Service. The Caregiving Transitions Study was further supported by an investigator-initiated grant (RF1 AG050609) from the NIA. Additional support was provided by the Johns Hopkins University Claude D. Pepper Older Americans Independence Center funded by the NIA (P30 AG021334). The content is solely the responsibility of the authors and does not necessarily represent the official views of the NINDS or the NIA.  Representatives of the NINDS were involved in the review of the manuscript but were not directly involved in the collection, management, analysis, or interpretation of the data.]

Additional Editor Comments:

For Fig. 1, you may move to supplemental data. 

Reviewers' comments:

Reviewer's Responses to Questions

**Comments to the Author**

1. Is the manuscript technically sound, and do the data support the conclusions?

Reviewer #1: Yes

2. Has the statistical analysis been performed appropriately and rigorously? 

Reviewer #1: Yes

3. Have the authors made all data underlying the findings in their manuscript fully available?

Reviewer #1: Yes

4. Is the manuscript presented in an intelligible fashion and written in standard English?

Reviewer #1: No

5. Review Comments to the Author

Reviewer #1: In this manuscript the relation between telomere length assessed on 2 occasions and becoming a caregiver or not in between the two assessments of telomere length are investigated.

Particularly the introduction and the methods section could use a spelling and grammar check. And also please make sure that you use the same font throughout the whole manuscript.

See more detailed queries below:

Introduction:

Please do not use citations from other papers, such as: "The authors stated that “Women with the highest levels of perceived stress have telomeres shorter on average by the equivalent of at least one decade of additional aging compared to low stress women" Please delete and paraphrase.

Why is the following relevant for this manuscript: "This publication has been cited over 3,000 times as of December 2021 according to Google Scholar, and has been widely shared on both family caregiver websites such as the Caregiver Action Network (25], and in foundation and news reports on family caregiving[26, 27]." Please delete.

In general in the introduction the authors provide too much details of every referred study. These methodological details would only be relevant if you are going to do it differently or that these methodological details explain why they ended up with certain results. Please focus only on the details that are relevant

I really miss a logical summary of the preceding literature on a more meta-level; I am not interested in all details, but I want to know what the results of preceding studies mean and to which extent they are comparable and if not why not and then – what it is that the current study will add to existing knowledge.

The aim of the current study seems overstated “These findings could address previous conflicting findings on this issue which is of great importance not only for our scientific understanding of the relationship between stress, telomeres, and health, but also for guiding our understanding of the pressing public policy and clinical issues faced as our aging population depends increasingly on family caregivers to provide care”

How so will you look at health and clinical issues related to aging? You will only look at telomere length and not something that is more clinically relevant. Moreover, as you show in the introduction this is hardly the first study to look at the relation between caregiving and telomere length.

Also please make clear whether you think these associations between caregiving and telomere length are specifically related to caregiving, or that caregiving is a form of chronic stress? And would this be the case for all types of caregiving, or not? In relation to stress (chronic stress, trauma, life events) several longitudinal studies with much longer follow-up and more assessments of telomere length have already been published (see for instance Gerritsen et al in Psychosomatic Medicine).

Be aware that you are using several fonts within one paragraph!

Methods

The description of the classification of whether a participant was a caregiver or not is confusing. Maybe this is due to the again alternating fonts, but possibly also because it seems as if you are repeating your self. Please rephrase lines 176-188.

Statistical analysis

Why include all variables that were used to match the control participants as covariates as well? The good thing of using a matched design is that you would not have to adjust for potential confounders as theoretically your cases and controls are highly comparable on those factors.

Figure 1 does not add to the results already shown in table 2. It also does not add new results to already existing data: we know there is a significant relation between telomere length at baseline and followup, similarly as there is with change between follow-up and baseline telomere length. The regression line is not interesting, as it is a general line and there are no separate lines for the caregivers and the controls. Which would also not lead to anything of interest, as there was no effect of caregiver status on telomere length. So I would suggest to delete this figure.

Discussion:

In the Discussion a large part is described about what other studies found with health-related outcomes. This is interesting, but I am missing how the authors think that health outcomes are related to telomere shortening. The link between health outcomes, stress and telomere length is currently missing, which in my view is crucial to make this relevant for the current study.

The scores on the perceived stress scale were very low, also for the caregivers. Please elaborate on how this could be and how this may have affected your results. Was this the same in other caregiver studies?

6. PLOS authors have the option to publish the peer review history of their article (what does this mean?). If published, this will include your full peer review and any attached files.

Reviewer #1: No

---

## [Author Response · Author response to Decision Letter 0]

15 Apr 2022

See uploaded Response to Reviewers file.

---

## [Decision Letter · Decision Letter 1]

6 May 2022

Telomere Shortening and the Transition to Family Caregiving in the Reasons for Geographic and Racial Differences in Stroke (REGARDS) Study

PONE-D-22-03368R1

Dear Dr. Roth,

We’re pleased to inform you that your manuscript has been judged scientifically suitable for publication and will be formally accepted for publication once it meets all outstanding technical requirements.

Kind regards,

Jian Gu

Academic Editor

PLOS ONE

Additional Editor Comments (optional):

Reviewers' comments:

Reviewer's Responses to Questions

**Comments to the Author**

1. If the authors have adequately addressed your comments raised in a previous round of review and you feel that this manuscript is now acceptable for publication, you may indicate that here to bypass the “Comments to the Author” section, enter your conflict of interest statement in the “Confidential to Editor” section, and submit your "Accept" recommendation.

Reviewer #2: All comments have been addressed

2. Is the manuscript technically sound, and do the data support the conclusions?

Reviewer #2: Yes

3. Has the statistical analysis been performed appropriately and rigorously? 

Reviewer #2: Yes

4. Have the authors made all data underlying the findings in their manuscript fully available?

Reviewer #2: Yes

5. Is the manuscript presented in an intelligible fashion and written in standard English?

Reviewer #2: Yes

6. Review Comments to the Author

Reviewer #2: (No Response)

7. PLOS authors have the option to publish the peer review history of their article (what does this mean?). If published, this will include your full peer review and any attached files.

Reviewer #2: No

---

## [Editor Report · Acceptance letter]

23 May 2022

PONE-D-22-03368R1 

Telomere shortening and the transition to family caregiving in the Reasons for Geographic and Racial Differences in Stroke (REGARDS) study. 

Dear Dr. Roth:

I'm pleased to inform you that your manuscript has been deemed suitable for publication in PLOS ONE. Congratulations! Your manuscript is now with our production department. 

Kind regards, 

on behalf of

Dr. Jian Gu 

Academic Editor

PLOS ONE